# Coping Strategies, Immediate and Delayed Suggestibility among Children and Adolescents

**Tiziana Maiorano and Monia Vagni ***

Department of Humanities, University of Urbino, 61029 Urbino, Italy; tiz.maiorano@gmail.com
* Correspondence: monia.vagni@uniurb.it; Tel.: +39-0722-305814

**Abstract:** Knowing the factors that influence children's suggestibility is important in implementing the psychological variables to be evaluated during a forensic evaluation. In the interrogative suggestibility model, coping strategies intervene in determining the acceptance or rejection of the leading question. However, studies that investigated the relationship between interrogative suggestibility and coping strategies had mixed results. Avoidance-oriented coping is associated with high level to immediate suggestibility and problem-focused with low levels. In this study, we measured immediate suggestibility, delayed suggestibility, and coping strategies in a sample of 100 children. We hypothesized that avoidance-oriented coping strategies have a predictive effect in increasing immediate suggestibility levels, in particular avoidance-oriented coping oriented towards the tendency to accept leading questions. No effect of coping strategies was expected on delayed suggestibility. All children completed the Gudjonsson Suggestibility Scale (GSS 2), a non-verbal IQ test, and the Coping Inventory for Stressful Situations (CISS). Coping strategies were not related to delayed suggestibility, but avoidance-oriented coping correlated positively with immediate suggestibility. Avoidance-oriented coping emerged as the only significant predictive model for shift and total suggestibility, and its subscale distraction emerged as a predictor for Yield 1 and Yield 2. No predictors emerged for delayed suggestibility. Results are discussed for their theoretical implications.

**Keywords:** suggestibility; coping strategies; GSS2; leading questions; children; adolescents

## 1. Introduction

The study of the mechanism of the suggestibility process in children and of the factors that influence the production of their responses is important to help research respond to the problem of children being interviewed in a forensic context (Vagni et al. 2018). Addressing this issue is important to inform forensic professionals in charge of interviewing children. Minors are often heard as witnesses by the police or judges, and it is therefore important to understand which cognitive processes they activate when faced with the leading questions and what strategies they use to reject or accept the suggestions. Suggestibility is a primary factor in determining a minor's ability to testify.

According to Schooler and Loftus (1993) there are two different mechanisms of suggestibility: immediate suggestibility, studied by the Gudjonsson and Clark (1986) model, and delayed suggestibility, typically studied by the misinformation effect paradigm. The first focuses on the immediate tendency to accept a leading question during an interview; the second focuses on the tendency to incorporate suggestive information in the memory.

A recent review highlights the demographic, cognitive, and psychosocial factors that affect children's suggestibility (Klemfuss and Olaguez 2020) but there is a lack of studies in the literature that analyze the role of coping strategies. On the other hand, there are studies that have used adult

samples with respect to both immediate suggestibility (Gudjonsson 1988; Forrester et al. 2001; Howard and Hong 2002; Bain et al. 2015) and delayed suggestibility (Zhu et al. 2010a, 2010b). Starting from the results of these studies we aimed to analyze the relationship between coping strategies and immediate and delayed suggestibility among children using an additional procedure for measuring suggestibility through the Gudjonsson Suggestibility Scale (Vagni et al. 2015; Gudjonsson et al. 2016) as used in several studies on samples of children (Vagni et al. 2017, 2018; Maiorano and Vagni 2019; Gudjonsson et al. 2020, 2021).

## 2. Literature Review

### 2.1. Interrogative Suggestibility: Gudjonsson and Clark

In Gudjonsson and Clark's model, interrogative suggestibility is defined as "the extent to which, within a closed social interaction, people come to accept messages communicated during formal questioning, as a result of which their subsequent behavioral response is affected'' (Gudjonsson and Clark 1986, p. 84). The model integrates two distinctive aspects of interrogative suggestibility: the impact of "leading questions" and "negative feedback".

The model aims at understanding the factors involved in the production of suggested answers, which imply the acceptance of leading questions or the modification of answers previously given following a negative criticism.

The premise behind the theory is that a witness or victim, in a context such as that of the interrogation, characterized by uncertainty and expectations, interacts with the interviewer and reacts to the questions that are posed according to his or her general cognitive system, adopting strategies to facing the stressful situation, which leads the witness or victim to engage in suggestive or refusing behavior when subjected to leading questions.

The forensic interview is a social situation that can generate stress reactions, uncertainty, and expectations of success in both adults and children (Gudjonsson 2003, 2018; Gudjonsson and Clark 1986; Vagni et al. 2015, 2018).

As previously stated, the model analyzes the possible strategies that a witness can implement to cope with pressing questioning according to their cognitive background and according to whether the feedback is negative or positive, rejecting or accepting the suggestion proposed (Gudjonsson 2013).

According to several authors (Ridley et al. 2013; Gudjonsson 2018; Drake et al. 2014) some factors are linked with suggestibility, such as age, intelligence quotient (Klemfuss and Olaguez 2020; Benedan et al. 2018; Gignac and Powell 2006; Gudjonsson and Henry 2003; Caprin et al. 2016; Chae and Ceci 2005; Roebers and Schneider 2001), self-esteem, confidence in one's memory, and even coping skills (Gudjonsson 1988). Yield could be more linked with cognitive factors and in particular with memory, intelligence, and language, while shift may therefore be more linked with anxiety and social and interpersonal factors (Gudjonsson 1984; Drake and Bull 2011; Ridley et al. 2013; Drake et al. 2014; Gudjonsson et al. 2016). The relationship between memory and suggestibility is complex and complicated. In particular, low memory scores in several studies have been associated with high immediate suggestibility scores and especially with the tendency to accept leading questions, whereas this relationship has not always been found with delayed suggestibility (Ridley and Gudjonsson 2013). In relation to age, significant differences have not always been found (Lee 2004). Furthermore, the link between suggestibility and age is greater in younger children (Brainerd et al. 2008). Moreover, there is contradictory evidence from suggestibility studies concerning age differences in middle childhood (Paz-Alonso and Goodman 2016).

Gudjonsson and Clark's model (1986; Gudjonsson 2018) begins by defining the social situation and the participants involved. The interrogative suggestibility is dependent upon the coping strategies that interviewees can use to cope and manage uncertainty and expectations of success experienced during the interview.

The model furthermore highlights the role of coping strategies, which are associated with the "General Cognitive Set" (Gudjonsson and Clark 1986), the general cognitive system that guides the

witness to assess the situation and decide to adopt general coping, which can facilitate suggestive or rejecting behavior towards the interviewing procedure. The interviewer asks a question that is subjected to a cognitive processing, which employs one or more general coping strategies to cope with the stressful situation. If the witness makes use of a strategy that includes a critical process of the situation, he/she will not be suggestible (Gudjonsson 2003). In the process of evaluation of each question, uncertainty and expectation intervene. Interviewees may be uncertain about answers to specific questions because they have a poor memory or no memory that will allow them to focus on the question.

The suggestibility is considered stable over time because it is linked to the cognitive abilities and to the personality characteristics.

This could suggest that both adults and minors tend in a stable way, or in different social contexts, to activate the same coping strategies to manage suggestive pressures. It becomes important, for the purposes of this study, to identify the main coping strategy used by a minor to predict its functioning during a witness interview by the police or a judge.

The role of coping strategies in the interrogative suggestibility model has been tested in previous studies using Gudjonsson Suggestibility Scales (Gudjonsson 1984, 1987), built to measure the tendency to yield to leading questions (yield) and the tendency to allow oneself to be influenced by the interviewer's negative feedback by modifying one's responses (shift).

## 2.2. Interrogative Suggestibility and Coping Strategies

Several studies that have tried to understand the relationship between interrogative suggestibility and coping strategies in order to verify whether the use of a strategy that includes a critical process of the situation predicts the ability to reject suggestions have obtained mixed results (Bain et al. 2015).

The first study (Gudjonsson 1988) focused on the relationship between coping strategies and interrogative suggestibility in order to test the theoretical hypotheses of the model and involved a sample of 30 adults who were administered the GSS1, Gudjonsson's Suggestibility Scale, and a subsequent interview about the strategies they had used during the test. The strategies, both cognitive and behavioral, were classified into active and avoidant, on the basis of the three categories described by Billings and Moos (1981):

(1) Active-cognitive methods that indicate the activation of attempts to manage thoughts and appraisal of the situation;
(2) Active-behavioral methods that include behavioral attempts to deal with the situation;
(3) Avoidance-oriented coping that indicates the avoidance of confrontation with the stressor.

The results showed that coping strategies could be highly predictive of suggestibility scores, since participants who resorted to an avoidant strategy were more suggestible than those who used an active coping strategy and who achieved higher yield scores (Yield 1 and Yield 2) to suggestive questions and negative feedback (shift).

According to Forrester et al. (2001), however, the study conducted by Gudjonsson (1988) had some notable limitations: the small number of participants, an unclear differentiation between coping strategies, ambiguous hypotheses, and the fact that the study started from the theoretical assumption that each participant would use only one strategy. In order to understand the relationship between interrogative suggestibility and the two types of "problem-focused" and "emotional" coping, Forrester et al. (2001) conducted a study on a sample of 61 students with an average age of 22.4 years (SD = 5.9), who were administered the Coping Orientation to Problem Experienced (COPE, Carver et al. 1989) and the GSS 1. Furthermore, the relationship between interrogative suggestibility and situational and dispositional coping was examined, with the aim of clarifying which was the best predictor. The results showed that the two coping styles were not significantly correlated with interrogative suggestibility.

Different results were obtained by Howard and Hong (2002) with a study conducted on a sample of 263 participants who were administered the COPE and GSS 1. Extracting from the initial

sample of 25 participants with emotional coping and 25 with problem-focused coping, a comparison between the two groups found that the more emotional participants scored higher in terms of yielding to suggestive questions (Yield 1) and total suggestibility score than those who used problem-focused coping. The results of this research reinforce the hypotheses regarding the dynamics and variables underlying the suggestibility process proposed by the Gudjonsson and Clark model.

In a more recent study conducted by Bain et al. (2015) in which 76 adults (45 males, 31 females), with a mean age of 31.37 (SD = 12.42), participated, the relationship between coping strategies, self-esteem, and interrogative suggestibility was examined. The authors hypothesized that problem-focused coping and self-esteem correlated negatively with suggestibility scores, while emotion-focused coping correlated positively. Participants were given the Gudjonsson Suggestibility Scale 2(GSS2), COPE, and the adult culture-free self-esteem questionnaire (CFSEI-2; Battle 1992). Significant models emerged for Yield 1, Yield 2, and total suggestibility in which emotion-focused coping was the only significant predictor. Correlational analysis and regression did not reveal any significant relationship between coping focused on emotions and the tendency to allow oneself to be influenced by the interviewer's criticism by modifying the initial responses following negative feedback (shift). Bain et al. (2015) argue that the results of this research confirm Gudjonsson and Clark's model (1986) and the hypothesis that suggestion responses are increased by the use of avoidance coping strategies.

In relation to delayed suggestibility, only one study has focused on the effect of coping strategies on a misinformation test, in which 436 university students participated. Zhu et al. (2010) found that false memories were positively correlated with active coping and negatively correlated with negative coping, unlike the results obtained by Gudjonsson (1988). According to Zhu et al. (2010) those who use an active style and are less fearful of negative evaluation may be overly confident in their answers even if based on the acceptance of misinformation but at the same time less likely to be influenced by questioning pressure.

Several studies have shown that avoidance coping strategy is likely to lead to higher immediate suggestibility levels, while the problem-focused coping strategy leads to a critical analysis of the situation and rejection of the suggestions (Gudjonsson 2018). In the interrogative suggestibility model, two main factors must be considered, intellectual abilities and memory skills, because they have an impact on the choice of the coping strategy. Intellectual and memory difficulties, such as learning disabilities or memory distrust syndrome, can increase distraction levels and compliance behavior, which increase suggestibility levels (Gudjonsson 1990, 1984, 1988). Borderline intelligence and learning disabilities are linked to higher suggestibility and normally lead to the choice of a less effective coping strategy (Gudjonsson 2018).

Avoidance coping strategy seems to lead the interviewee to give answers considered more plausible with the external cues provided instead of activating a critical evaluation. Non-suggestible coping strategy involves a critical analysis of the questions and improves the problem-solving solution (Howard and Hong 2002; Gudjonsson 2018).

There are no existing studies on the effects of coping strategies on delayed suggestibility following the model of interrogative suggestibility.

### 2.3. Coping Strategies

According to the conceptualization of coping proposed by Endler and Parker (1990, 1999), in addition to the fundamental modalities of coping, described in previous theories, emotion-focused coping (thoughts and behaviors that a person uses to regulate discomfort) and problem-focused coping (attempts made to manage the problem causing the discomfort), also used in studies to detect the relationship between coping and interrogative suggestibility (Gudjonsson and Clark 1986), there is a third type: avoidance-oriented coping, either task-oriented (e.g., distracting oneself with another task) or person-oriented (e.g., social diversion, such as seeking out others), which generally involves distancing oneself from the stressful situation.

Avoidance-oriented coping includes both strategies aimed at the person and at the maneuver, which involve the attempt to avoid the problem and relieve stress. Person-oriented avoidance attempts consist of seeking social diversions, while task-oriented avoidance strategies involve striving for cognitive changes, achieved through distraction and undertaking surrogate tasks. Starting from this theoretical perspective of coping, Endler and Parker (1990, 1994, 1999) built the Coping Inventory for Stressful Situations (CISS), which presents a form for adults and a form for adolescents (Sirigatti and Stefanile 2009).

## 3. Objectives

The aim of this study was to explore the relationship between age, memory, coping styles, and immediate and delayed suggestibility in children and adolescents. A leading question may be considered a critical event because the subject displays a situation of uncertainty by providing incorrect information. In a critical situation, people inevitably activate coping strategies to answer questions and to manage negative criticism and the expectation of success. If, as in the case of the Gudjonsson and Clark model (1986) of interrogative suggestibility, the interviewee is exposed to criticism about his or her performance, in order to cope with this stressor and satisfy the expectation of success, he/she activates coping strategies.

In this sense, problem-focused and task-oriented coping strategies could be associated with greater resistance to interrogative suggestibility because there is a critical attitude towards the event (leading question). On the contrary, those who tend to resort to avoidance strategies and in particular the task-oriented avoidance strategy (distracting oneself with another task), when faced with leading questions, will adopt a non-critical strategy for solving the problem and thus tend to present higher levels of suggestibility.

Previous studies have verified the predictive effect of coping strategies on the levels of immediate suggestibility of adults. The present study aimed to verify the predictive effect of coping strategies on both immediate and delayed suggestibility of children.

Starting from these assumptions, this study tested the following hypotheses:

**Hypothesis 1.** *Coping styles are associated with age, immediate recall, and immediate suggestibility.*

**Hypothesis 2**. *Avoidance-oriented coping strategies, age, and immediate recall have a predictive effect on immediate suggestibility levels.*

**Hypothesis 3**. *Delayed suggestibility is not associated with coping strategies because it is linked to factors that are different and independent from those involved in the process of answering leading questions.*

## 4. Materials and Methods

### 4.1. Participants

The study was carried out on 100 children and adolescents: 56 males aged between 10 and 14 years (M = 12.02, SD = 1.104) and 44 females, aged between 10 and 15 years (M = 12.16, SD = 1.010).

The ecological sample was selected randomly from several Italian schools. We considered only the socio-demographic variables recognized by the literature as significant predictors of suggestibility: age, gender, and intelligence quotient (IQ; Hritz et al. 2015; Ridley et al. 2013).

Following the literature, children with cognitive disabilities and/or with proven intellectual retardation or learning disabilities were excluded from the sample.

Raven's Matrices (Raven et al. 1998) were administered to participants for testing non-verbal IQ and for screening those who had out-of-average IQ scores. According to literature, IQ affects suggestibility scores and for this reason was tested in the study. The IQ of all participants fell within the average range of 90–110 (M= 103.2, SD= 6.87).

*4.2. Procedure*

The instruments were administered following the same procedure with all the participants.

The tendency to immediate suggestibility was measured by the administration of the GSS2, following the standard procedure (Gudjonsson 1997). Participants were asked to listen to the reading of a story and to provide their recollection of the events in the story immediately afterwards. The instrument then provided an intervening period of about 50 min before administration of the GSS2 interview. During this period, both Raven's Matrices was administered, both as a distracting task and as a measure of IQ, as well as the Coping Inventory for Stressful Situations (CISS). After administration of these tests, the interview provided by the GSS2 was conducted to obtain the scores for the Yield 1, Yield 2, shift, and total suggestibility scales.

After 1 week, participants were asked to tell the story again to check delayed recall and delayed suggestibility, as indicated in the additional procedure (Gudjonsson et al. 2016; Vagni et al. 2015) used in several studies (Vagni et al. 2017, 2018; Wachi et al. 2019).

The GSS2 was administered individually. The materials were used with the authorization of the parents or guardians of the minors involved, in accordance with the Declaration of Helsinki. The institutional ethics committee approved the procedure.

*4.3. Instruments*

The Italian version of the Gudjonsson Suggestibility Scale 2 (GSS2, Gudjonsson 1997) instrument was used to detect memory accuracy and the tendency to interrogative suggestibility. This scale has been revisited and applied in previous studies (Gudjonsson et al. 2016; Vagni et al. 2015). This version was applied to a sample of 1183 minors, with Cronbach's alpha coefficients having values commensurate with good reliability: Yield 1, $\alpha = 0.81$; Yield 2, $\alpha = 0.83$; shift, $\alpha = 0.71$; and total suggestibility, $\alpha = 77$ (Gudjonsson et al. 2016). The following reliability coefficients were calculated for the sample of this study: 0.80, 0.86, 0.78, and 0.84 for Yield 1, Yield 2, shift, and total suggestibility, respectively.

The GSS2 consists of reading a short story and asking for an immediate recall. After 50 min, 20 questions, 15 of which are misleading, are administered. At the end of the first interview, negative feedback is given and then the 20 questions are asked again.

The scale can obtain immediate recall as memory score (the number of elements of the target stimulus recalled immediately after reading) and the following immediate suggestibility scores:

Yield 1 measures the number of suggestions accepted in response to leading questions in the first interview (the maximum score being 15).

Yield 2 measures the number of suggestions accepted in response to leading questions in the second interview, after negative feedback (the maximum score being 15).

Shift corresponds to the number of times the participants change their answers after negative feedback (the maximum score being 20).

Total suggestibility corresponds to the sum of Yield 1 and shift (the maximum score being 35).

The agentive procedure of this instrument (Vagni et al. 2015; Gudjonsson et al. 2016), which has been used in several studies on samples of minors (Gudjonsson et al. 2020, 2021), was administered to measure delayed recall and delayed suggestibility scores.

Delayed recall is the number of elements of the target stimulus recalled after 1 week.

Delayed suggestibility is the number of suggestions accepted and included in the delayed recall.

The Coping Inventory for Stressful Situations (CISS, Endler and Parker 1990; Sirigatti and Stefanile 2009) is an instrument that evaluates coping strategies in children and has 3 scales: task-oriented (T), emotion-oriented (E), and avoidance (A). The task-oriented strategy measures the tendency to use concrete problem solving, in order to cognitively reconstruct the problem or attempt to alter the situation and re-plan behavior. Emotion-oriented strategy describes emotional reactions that are self-oriented. This coping strategy aims at reducing stress but is not always successful. Reactions include anger, self-preoccupation, blame. This scale was associated with high levels of stress (Endler and Parker 1990). Avoidance describes activities and cognitive changes aimed at avoiding the stressful situation. This can occur via distracting oneself with other situations

or tasks or via social diversion. There are two sub-scales for the avoidance-oriented strategy scale: distraction and social diversion. Distraction (D) indicates the tendency to avoid stressful situations through distraction with other tasks (Task-oriented) or engaging in social activities (social diversion). This instrument is composed of 48 items evaluated on a five-point scale (1 = never; 5 = very often). The alpha coefficients of the Italian version are the following for each scale: task-oriented ($\alpha$= 0.86), emotion-oriented ($\alpha$= 0.84), avoidance ($\alpha$= 0.86), and for subscales of avoidance they are 0.76 and 0.79 for distraction and social diversion, respectively. In this study, because the CISS in the original version can be administered to children aged 13 years upwards and the Italian standardization is for children aged 16 years upwards, raw scores were used for analysis. Moreover, alpha coefficients were calculated for this sample: task-oriented ($\alpha$= 0.88), emotion-oriented ($\alpha$= 0.78), avoidance ($\alpha$= 0.82), and for the subscales of avoidance they were 0.76 and 0.75 for distraction and social diversion, respectively.

*4.4. Analytics Strategy*

In the present study a preliminary analysis was conducted using Pearson's correlational analysis in order to estimate the relationship between suggestibility scores, coping strategies, and other variables such as age and memory scores. In order to test the central hypotheses, multiple regression analyses were conducted to examine the predictive power of coping strategies on suggestibility scores.

## 5. Results

Descriptive statistics associated with the GSS 2 and CISS subscales scores are shown in Table 1.

**Table 1.** Average Age and Scores for the GSS 2 and CISS (*n* = 100).

| Explanatory Variable | Min | Max | Mean | SD |
|---|---|---|---|---|
| Age | 10 | 15 | 12.08 | 1.061 |
| GSS2 | | | | |
| Immediate Recall | 6 | 32 | 16.85 | 5.01 |
| Delayed Recall | 0 | 27 | 13.96 | 5.802 |
| Yield 1 | 0 | 13 | 6.68 | 3.306 |
| Yield 2 | 0 | 15 | 8.06 | 4.032 |
| Shift | 0 | 13 | 5 | 3.005 |
| Total | 1 | 23 | 11.68 | 5.197 |
| Delayed Suggestibility | 0 | 5 | 0.72 | 1.043 |
| CISS | | | | |
| Task-Oriented | 17 | 76 | 47.47 | 12.172 |
| Emotion-Oriented | 17 | 69 | 38.51 | 9.784 |
| Avoidance | 20 | 88 | 42.5 | 12.792 |

Note: GSS2 = Gudjonsson Suggestibility Scale 2; CISS = Coping Inventory for Stressful Situations.

Before conducting the main analysis, a preliminary analysis was conducted using Pearson's correlations between GSS2 immediate recall and suggestibility scores, CISS Scores, and age in order to study the relationship between these variables. Table 2 shows the results of correlations.

**Table 2.** Pearson's correlation between the GSS2 and CISS ($n$ = 100).

| Explanatory Variable | Age | GSS2 | | CISS | | | |
|---|---|---|---|---|---|---|---|
| | | Immediate Recall | Task-Oriented | Emotion-Oriented | Avoidance | Avoidance Subscales | |
| | | | | | | Distraction | Social Diversion |
| Age | 1 | −0.27 | 0.327 ** | 0.333 ** | 0.238 ** | 0.149 | 0.211 * |
| GSS2 Scores | | | | | | | |
| Immediate Recall | 0.03 | 1 | −0.003 | 0.008 | −0.173 * | −0.186 * | −0.018 |
| Delayed Recall | 0.009 | 0.74 ** | −0.018 | −0.147 | −0.167 | −0.165 | 0.004 |
| Yield 1 | −0.076 | −0.36 ** | −0.061 | −0.031 | 0.220 * | 0.267 ** | 0.083 |
| Yield 2 | −0.012 | −0.235 ** | 0.014 | −0.073 | 0.184 * | 0.233 ** | 0.047 |
| Shift | −0.02 | −0.15 | 0.11 | 0.011 | 0.240 ** | 0.223 * | 0.186 * |
| Total Suggestibility | −0.06 | −0.316 | 0.024 | −0.013 | 0.279 ** | 0.299 ** | 0.16 |
| Delayed Suggestibility | −0.08 | −0.167 | 0.003 | −0.121 | −0.011 | −0.015 | 0.039 |

Note: * $p < 0.05$, ** $p < 0.01$; GSS2 = Gudjonsson Suggestibility Scale 2; CISS = Coping Inventory for Stressful Situations.

Only avoidance-oriented coping correlates significantly with immediate recall (r = −0.17) and immediate suggestibility: Yield 1 (r = 0.22), Yield 2 (r = 0.18), shift (r = 0.24), and total suggestibility (r = 0.28). It did not correlate significantly with delayed recall and delayed suggestibility. Its subscale distraction correlates significantly with immediate recall (r = −0.019), Yield 1 (r = 0.27), Yield 2 (r = 0.23), shift (r = 0.22), and total suggestibility (r = 0.30). In particular, immediate recall shows a small significant negative correlation with avoidance and the distraction subscale; Yield 1 and Yield 2 shows moderate positive correlation with distraction, while shift shows significant positive correlation with avoidance. This indicates that participants who more frequently use an avoidance strategy are more suggestible to leading questions, and in particular a moderate correlation can be seen between distraction (avoidance-oriented coping responses task-oriented) and yield, which is the component of immediate suggestibility that is most influenced by cognitive factors (Ridley and Gudjonsson 2013). In relation to age, no significant correlations are evident with GSS2 scores, in agreement with Lee (2004), while there are significant positive moderate correlations between age and coping strategies

On the basis of the results of the correlation analysis, multiple regression analyses were conducted to examine the predictive power of the avoidance scale and its subscale distraction in explaining the variance in GSS2 scores, including age and immediate recall.

Distraction was assumed to be a predictor of Yield 1 and 2 in consideration of the higher correlation with respect to avoidance, while for shift the predictor was assumed to be avoidance because only small correlations emerged in the subscales. For Yield 1, distraction strategy (β= 0.224, *t* = 2.349, *p* = 0.02) and immediate recall (β = −0.31; *t* = −3.35; *p* = 0.001) emerged as predictors, and the model explained 18% of variance. For Yield 2, distraction strategy (β = 0.22, *t* = 2.218, *p* = 0.029) emerged as the only significant predictor, and the model explained 11% of variance. For total suggestibility, distraction strategy (β = −0.26; *t* = 2.75; *p* = 0.007) and immediate recall (β = −0.27; *t* = −2.79; *p* = 0.006) emerged as predictors, and the model explained 17% of variance. In step 2, the effect of immediate recall is reduced by distraction strategy (step 1: β = −0.32; *t* = −3.27; *p* = 0.001; see Table A1).

Table A2 shows the predictive effects of the avoidance strategy on GSS2 score. For shift, avoidance strategy (β = 0.24, t = 2.233, p = 0.02) emerged as the only significant predictor, and the model explained 8% of variance. For total suggestibility, avoidance strategy (β = 0.26, *t* = 2.66, p = 0.009) and immediate recall (β = −0.268, *t* = 2.821, p = 0.006) emerged as predictors, and the model explained 16% of variance. In the models, age did not emerge as a predictor, and no differences emerged between adolescents and children in the avoidance strategy scores. No significant model emerged for delayed suggestibility.

All models are reported in Appendix A.

## 6. Discussion

The main hypothesis of this study was that the use of problem-focused coping is a protective factor in stressful situations such as a suggestive interrogation, while the use of avoidance coping strategies are predictive of high suggestibility scores. This hypothesis was partially confirmed; avoidance-oriented coping emerged as the only coping strategy that explains the immediate suggestibility score, as previously found by Gudjonsson (1988). In particular, the task-oriented distraction subscale of avoidance affects yield scores, which are the components of immediate suggestibility that are most influenced by cognitive factors (Ridley and Gudjonsson 2013).

Normally, as recognized in other studies (Ridley and Gudjonsson 2013; Gudjonsson et al. 2016), high scores for immediate recall can reduce vulnerability to suggestive questions, especially on Yield 1. Yield 1 is the suggestibility component that is most associated with cognitive factors such as memory, intelligence, and language (Gudjonsson and Young 2010; Gudjonsson and Henry 2003; Bruck and Melnyk 2004; Benedan et al. 2018). Previous studies have focused on the predictive effect of coping strategies on the levels of immediate suggestibility of adults. This study, on the contrary, involved children and aimed to verify the predictive effect of coping strategies on both immediate and delayed suggestibility.

The results shown in this study highlight how the use of distraction and avoidance strategies reduces the protective effect of immediate recall and increases immediate suggestibility. The results seem to indicate that in both children and adults the use of avoidance coping strategies leads to an increase of the immediate suggestibility levels. According to several studies, avoidance coping strategies particularly increase both the shift and total suggestibility scores detected after the negative feedback. (Gudjonsson 1988, 2018; Bain et al. 2015). Avoidance coping strategy seems to link to social and interpersonal pressure factors.

The correlational analysis and the regression models of this study seem to highlight that after negative feedback there is a greater effect of the avoidance strategies on the immediate suggestibility scores. Moreover, the study highlighted how age does not affect suggestibility as found in other studies (Lee 2004; Ridley and Gudjonsson 2013). These data may be due to the fact that age has a greater impact on suggestibility in younger children, while our sample, which included children in middle childhood and adolescents, showed average suggestibility scores. Although age is related to the use of coping strategies, it does not significantly affect the use of the avoidance strategy that raises the levels of suggestibility regardless of age.

This seems to suggest that those who during the first interview (Yield 1) made use of avoidance strategies, in response after feedback to the second suggestive interview (Yield 2), increased their avoidance strategy and especially distraction. The results obtained in this study, however, seem to highlight how the distraction subscale, which indicates the tendency to focus the attention on other cognitive tasks, has a predictive effect on Yield 1 and 2. These effects confirm the theoretical implications of the interrogative suggestibility model according to which yield scores are linked to cognitive factors (according to Gudjonsson 2003). The present study confirms, in accordance with Gudjonsson 1988, that the use of an avoidance coping strategy increases the susceptibility to suggestions but uniquely highlights how the tendency to use distraction to avoid stress through the task-oriented avoidance strategy, which represents a cognitive strategy in particular, affects the cognitive components of the suggestibility.

According to Bain et al. (2015) task-oriented coping strategy (a problem-focused activity) was not a predictor. Moreover, delayed suggestibility is not affected by coping strategy, confirming that immediate and delayed suggestibility are two different mechanisms (Ridley and Gudjonsson 2013). This study confirms that the relationships between coping strategies and suggestibility are complex and in agreement with other studies that have given mixed results.

According to several studies carried out on adults, it is very important to evaluate the immediate and delayed suggestive vulnerability of the children and their main coping strategies in the legal context (Ridley et al. 2013; Gudjonsson 2018). The type of coping strategy utilized by children facing leading questions can predict their ability to reject and resist the pressures involved during a forensic interview. This study presents some practical implications in the forensic field because witness interview conditions involve a high level of stress. It is therefore important to understand how children manage stress so that they can give their best performance. Children who show difficulty tolerating stress and frustration at the judge's lack of understanding of their responses may resort to greater use of the avoidance and distraction strategies with the risk of being more suggestible. In these cases, the expert could suggest interviews favoring as much as possible spontaneous narration, shorter interview times, and not addressing repetition of the questions and favoring coping strategies that are more oriented to the task. In other words, detecting the dominant coping strategy of minor witnesses before listening to them may allow the structuring of a more adequate interview, leading to a better performance on the part of the witness and avoiding the risks of suggestibility factors.

This study had some limitations: the sample was small and did not allow generalization of the results. A self-report questionnaire was used for measuring coping strategies, and this may have led to difficulty in understanding some items, particularly in younger children.

In future study, measures of coping styles specific to interrogative situations could be used in order to analyze strategies used for coping during interrogative performance, with the adoption of an instrument built ad hoc. Furthermore, recent research has focused on the types of response to

Gudjonsson's Suggestibility Scale (Gudjonsson et al. 2021). The authors highlighted the importance not only of the number of times the interviewee accepts or rejects a suggestion but also the type of response: "no"; "don't know" and "not mentioned", and thus in future study the role of coping in this paradigm could be considered in order to verify if avoidance coping influences the choice of the "don't know" answer and at same time if problem-focused coping strategy can increase the frequency of the "not mentioned" answer being given to leading questions.

## 7. Conclusions

Although this research has limits, it has the originality of having studied for the first time the relationship between coping and suggestibility strategies in children and is also the only study to have investigated the effect of this relationship on delayed suggestibility. It also adds important information both about the suggestibility of children and about how children who use avoidance strategies and in particular who tend to distract themselves from the task in order to lower the stress level of the event are more suggestible. These findings present an important development for experts in forensic situations and could help in assessing the extent to which children are able to cope with stressful situations, in particular when subjected to judicial interview. In order to develop this study of the role of coping strategies in decision-making processes in suggestive situations, it would be interesting in future research to study not only child witnesses' simple yes/no answers to leading questions but also to examine the type of response they give in terms of certainty, doubt, and memory distrust and investigate their association with coping strategies in order to test their efficacy.

**Author Contributions:** Both authors contributed equally to the conceptualization, methodology, validation, formal analysis, investigation, data curation, writing—original draft preparation, writing—review and editing, All authors have read and agreed to the published version of the manuscript. All authors have read and agreed to the published version of the manuscript.

**Funding:** This research received no external funding.

**Conflicts of Interest:** The authors declare no conflict of interest.

## Appendix A

**Table A1.** Hierarchical linear regression models for Yield 1 and Yield 2 scores ($n = 100$).

| Explanatory Variable | Yield 1 | | Yield 2 | |
|:---:|:---:|:---:|:---:|:---:|
| | Ex (B) | B | Ex (B) | B |
| **Step1** | | | | |
| Immediate Recall | −0.236 | −0.358 *** | -0.186 | −0.232 * |
| Age | −0.207 | −0.67 | −0.43 | −0.113 |
| | $R^2 = 0.134$ | | $R^2 = 0.068$ | |
| | F = 7.049 *** | | F = 3.52 * | |
| **Step 2** | | | | |
| Immediate Recall | −0.208 | −0.315 *** | −0.153 | −0.19 |
| Age | −0.314 | −0.101 | −0.558 | −0.147 |
| Distraction | 0.11 | 0.224 * | 0.132 | 0.22 * |
| | $R^2 = 0.181$ | | $R^2 = 0.113$ | |
| | $\Delta R^2 = 0.47$ * | | $\Delta R^2 = 0.45$ * | |
| | F = 7.068 *** | | F = 4.086 ** | |

Note: * $p < 0.05$, ** $p < 0.01$, *** $p < 0.001$; distraction= task-oriented avoidance subscale − CISS scale.

**Table A2.** Hierarchical linear regression models for shift, total suggestibility, and delayed suggestibility (*n* = 100).

| Explanatory Variable | Shift | | Total Suggestibility | | Delayed Suggestibility | |
|---|---|---|---|---|---|---|
| | Ex (B) | B | Ex (B) | B | Ex (B) | B |
| **Step1** | | | | | | |
| Immediate Recall | −0.90 | −0.15 | −0.326 | −0.315 *** | −0.03 | −0.016 |
| Age | −0.51 | −0.18 | −0.259 | −0.53 | −0.07 | −0.075 |
| | $R^2 = 0.023$ | | $R^2 = 0.103$ | | $R^2 = 0.034$ | |
| | F = 1.15 ns | | F = 5.52 ** | | F = 1.62 ns | |
| **Step 2** | | | | | | |
| Immediate Recall | −0.06 | −0.107 | −0.278 | −0.268 ** | −0.03 | −0.017 |
| Age | −0.217 | −0.076 | −0.568 | −0.11 | −0.069 | −0.069 |
| Avoidance | 0.56 | 0.24 * | 0.10 | 0.26 ** | −0.02 | −0.028 |
| | $R^2 = 0.075$ | | $R^2 = 0.164$ | | $R^2 = 0.034$ | |
| | $\Delta R^2 = 0.052$ * | | $\Delta R^2 = 0.061$ ** | | $\Delta R^2$ = ns | |
| | F = 2.61 * | | F = 6.28 *** | | F = 1.09 ns | |

*Note: * $p < 0.05$, ** $p < 0.01$, *** $p < 0.001$; avoidance-oriented coping = CISS scale.*

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
