# Peer review of "Coping Strategies, Immediate and Delayed Suggestibility among Children and Adolescents"

_socsci, doi:10.3390/socsci9110186_

Round 1

Reviewer 1 Report

Overall, this is an interesting study and well-written manuscript that has the potential to shed more light on the relationship between age, memory, coping styles, and immediate and delayed suggestibility in children and adolescents. The introduction and discussion were suitable and wide, large literature was referenced. However, some points require further attention (see in pdf file).

Author Response

COVER LETTER Reviewer 1

The authors thank the reviewer for useful suggestions which have all been accepted and included in the manuscript as specified below.

  1. Objectives should be move as separate section (Line 193):

REPLY: We have proceeded to move “Objectives” as a separate section and have changed the order of paragraphs.

  1. The author should be change aim to : The aim of this study was to explore the relationship between age, memory, coping styles and immediate and delayed suggestibility in children and adolescents.

REPLY: We have changed aim as suggested. In agreement with the reviewer we think this is more appropriate.

3.Hypothesesshould be changed. The authors should write about the relationship between more factors (eg. Age, memory, and coping styles) and immediate and delayed suggetsibility.

REPLY: The Hypotheses have been changed.

1) Hypothesis 1: Coping styles are associated with age, immediate recall and immediate suggestibility

2)        Hypothesis 2: Avoidance-oriented coping strategies, age and immediate recall have a predictive effect on immediate suggestibility levels

4.The term Intellectual quotient should be changed to “Intelligence quotient”

REPLY: We apologise for the mistake. The word has been changed

  1. The authors should write better operationalization of the variables. What is the indicator of the immediate suggestibility? What is the indicator of the delayed suggestibility?

REPLY: We have better explained the variables as follows:

The GSS2 consists in reading a short story and asking for an immediate recall. After 50 minutes 20 questions, 15 of which are misleading, are administered. At the end of the first interview a negative feedback is given and then the 20 questions are asked again

The scale allows the evaluation of immediate recall as memory score (the number of elements of the target stimulus recalled immediately after reading). It also allows the following immediate suggestibility scores:

Yield1measures the number of responses accepted to leading questions (the maximum score being 15).

Yield 2 measures the number of responses accepted to leading questions in the second interview, after negative feedback (the maximum score being 15).

Shift corresponds to the number of times the participants change their answers after negative feedback (the maximum score being 20).

Total Suggestibility corresponds to the sum of Yield 1 and Shift (the maximum score being 35).

The further procedure of this instrument (Vagni et al. 2015; Gudjonsson et al. 2016) which has been used in several studies on samples of minors (Gudjonsson et al. 2020; Gudjonsson et al. 2021) was administered to measure delayed recall and delayed suggestibility scores.

  1. The results should be changed. Firstly, the authors should present the Table with a matrix correlation between independent variables (age, memory, coping styles) and dependent variables (immediate and delayed suggestibility). Secondly, if there is a significant correlation they should present Tables with analysis of regression. In my opinion, there is no reason to present a comparison between females and males or children and adolescents, because the authors used age as an independent variable (factor).

REPLY: In agreement with the reviewer Pearson’s correlations between the independent variables (age, memory, and coping styles) and dependent variables have been added. The results were discussed following the literature that was cited in the manuscript. There is no correlation between age and GSS2,but in agreement with the literature age was kept in the regression analysis because it is in any case a control factor on suggestibility scores. Age shows significant correlations with coping styles. Immediate recall shows correlations both with GSS2 scores and coping styles.

The comparisons between gender and age groups have been removed.

Reviewer 2 Report

The paper addresses an interesting question about the relationship between coping strategies and suggestibility using a sample of Italian children and adolescents. In general, the paper is well written and the literature review is thoughtful. However, some shortcomings should be considered to improve the paper’s quality:

  • The authors should present validation data concerning the Italian version of the GSS 2;
  • The authors should explain what was the age cut–off point to divide children from adolescents since this issue might be relevant to ascertain age as moderator variable;
  • For the CISS it would be important to present Cronbach’s Alpha obtained in the validation study;
  • Also it would be also relevant to produce Cronbach’s Alpha values for GSS 2 and CISS in the present study;
  • On the discussion, the authors often refer to other studies or literature without presenting any reference (see lines 368/369, 391 and 395);
  • Some wording needs correction (see for example lines 226-227 and 368-370);
  • The conclusion deserves improvement namely regarding on how to develop future research in the current topic as suggested in the last paragraph of the discussion.

Author Response

COVER LETTER REVIEWER 2

The authors thank the reviewer for the useful suggestions which have all been accepted and included in the manuscript as specified below.

  • The authors should present validation data concerning the Italian version of the GSS 2;

REPLY: In agreement with the reviewer the validation data of the Italian version of the GSS2 have been added.

  • The authors should explain what was the age cut–off point to divide children from adolescents since this issue might be relevant to ascertain age as moderator variable;

REPLY: In agreement with other reviewer the dichotomic age variable has been removed and age considered as a continuous variable. We have added correlations between age and immediate recall with GSS2 and coping styles scores (see table2).

  • For the CISS it would be important to present Cronbach’s Alpha obtained in the validation study;

REPLY: Cronbach’s Alpha of the Italian validation have been added to the manuscript.

  • Also it would be also relevant to produce Cronbach’s Alpha values for GSS 2 and CISS in the present study;

REPLY: Cronbach’s Alpha values for GSS2 and CISS of this study have been included. All coefficients can be considered adequate and satisfactory

  • On the discussion, the author soft enrefer to other studies or literature without presenting any reference (see lines 368/369, 391 and 395);

REPLY: The authors apologize for these oversights and have specified the citations in the lines indicated

  • Some wording needs correction (see for example lines 226-227 and 368-370);

REPLY: We apologise for these errors. The corrections have been made.

The conclusion deserves improvement namely regarding on how to develop future research in the current topic as suggested in the last paragraph of the discussion.

REPLY: This suggestion is very helpful. In the conclusion we have explained the possible future developments of this study. Some comments on the application or practical implications of the results obtained have also been added in the discussion.